# Expanding continual few-shot learning benchmarks to include recognition of specific instances

**Gideon Kowadlo** [ID]*, **Abdelrahman Ahmed, Amir Mayan, David Rawlinson**

Cerenaut, Melbourne, Australia

* gideon@cerenaut.ai

**Data Availability Statement:** The datasets are freely available standard computer vision datasets (Omniglot and Slimagenet64). The source code is open source: https://github.com/Cerenaut/cfsl.

## Abstract

Continual learning and few-shot learning are important frontiers in progress toward broader Machine Learning (ML) capabilities. Recently, there has been intense interest in combining both. One of the first examples to do so was the Continual few-shot Learning (CFSL) framework of Antoniou et al. (2020). In this study, we extend CFSL in two ways that capture a broader range of challenges, important for intelligent agent behaviour in real-world conditions. First, we increased the number of classes by an order of magnitude, making the results more comparable to standard continual learning experiments. Second, we introduced an 'instance test' which requires recognition of specific instances of classes—a capability of animal cognition that is usually neglected in ML. For an initial exploration of ML model performance under these conditions, we selected representative baseline models from the original CFSL work and added a model variant with replay. As expected, learning more classes is more difficult than the original CFSL experiments, and interestingly, the way in which image instances and classes are presented affects classification performance. Surprisingly, accuracy in the baseline instance test is comparable to other classification tasks, but poor given significant occlusion and noise. The use of replay for consolidation substantially improves performance for both types of tasks, but particularly for the instance test.

## Introduction

Over the past decade, Machine Learning (ML) has made enormous progress in many areas. Typically, a model learns from a large iid dataset with many samples per class, and after a training phase, the weights are fixed i.e., it does not continue to learn. This is limiting for many applications, and as a result, distinct subfields have emerged that embrace different learning requirements, such as continual learning and few-shot learning.

### Continual learning

In many real-world applications, new data are continually introduced—they are not all available for an initial training phase as assumed in typical ML settings. Continual learning (or life-long learning) is the ability to learn new tasks while maintaining performance on previous

**Funding:** This initiative was partly funded by the Department of Defence and the Office of National Intelligence under the AI for Decision Making Program, delivered in partnership with the Defence Science Institute in Victoria. The funders did not play any role in the study design, data collection and analysis, decision to publish or preparation of the manuscript.

**Competing interests:** The authors have declared that no competing interests exist.

tasks. A well-known difficulty is catastrophic forgetting [1], which recognises that new learning with different data statistics disrupts existing knowledge. There are many approaches to mitigate catastrophic forgetting that fall broadly into 3 categories [2]: Regularisation-based methods, which share representational responsibility between model parameters, Parameter isolation methods, which only modify a subset of parameters in response to new data, and Replay methods, which are inspired by Hippocampal replay [3] and allow models to continue to learn slowly by internally repeating samples according to different policies.

## Few-shot learning

Another major challenge common in real-world problems, is the fact that there may not be many samples to learn from. In few-shot learning, only a few samples of each class are available. In the standard framework [4, 5], background knowledge is first acquired in a pretraining phase with many classes. Then, one or a few examples of a novel class are presented for learning, and the task is to identify this class in a test set (typically 5 or 20 samples of different classes). Knowledge of novel classes is not permanently integrated into the network, which precludes continual learning.

## Continual few-shot learning

*Continual few-shot learning* is also known as *few-shot continual learning*. Realistic environments do not keep continual and few-shot learning neatly separated. Effective agents should be capable of both of them simultaneously—an enviable characteristic of human and animal learning. We need to accumulate knowledge quickly and may only ever receive a few examples to learn from. For example, given knowledge of vehicles (e.g. trucks, cars, bikes etc.), we can learn about any number of additional novel vehicles (e.g. motorbike, then skateboard) from only a few examples. CFSL is critical for everyday life, particularly artificial agents in dynamic environments and many industry applications. For example, a pick-and-place robot should be able to recognise and handle new products after being shown just one demonstration. Furthermore, reasoning about specific instances is also important. For example, we may require the pick-and-place robot to put all the cereal boxes into a *specific* bin.

## Establishing benchmarks in continual and few-shot learning

While there are several established benchmarks in Continual learning and Few-shot learning individually, consensus regarding appropriate Continual *and* Few-shot learning benchmarks is still emerging [6]. Defining benchmarks is crucial for effective research progress, because benchmark conditions and characteristics strongly affect the potential performance of various models and algorithms. Frustratingly, many methods are only applied to one benchmark contender, making results incomparable to algorithms or models applied to alternative benchmarks.

The aim of this work is to broaden the range of capabilities covered in CFSL benchmarks to include recognition of specific instances of an object, regardless of class. Recognition of specific instances is an important capability that is routine for humans and other animals but is largely neglected by ML. For example, you usually know which coffee cup is yours, even if it appears similar to the cup of tea that belongs to your colleague. It is easy to see how this capability has applications across domains from autonomous robotics, to dialogue with humans or applications such as fraud detection. Reasoning about specific instances underpins memory for singular facts and an individual's own autobiographical history [7], and is therefore important for decision making and planning.

Recognition of specific instances is a special case of few-shot learning and is not equivalent to one-shot learning of classes. Learning specific instances requires the ability to learn the distinct characteristics of a specific instance of a class, differentiate between very similar samples, and differentiate samples of the same class. These capabilities imply memorisation, but simplistic memorisation strategies will not be invariant to changing observational conditions such as object or sensor pose, occlusion, and measurement noise.

In this paper, we extended the CFSL benchmark introduced by Antoniou et al. [6] with: a) an instance test, as well as b) an increase in scale of the existing experiments to be comparable to other continual learning studies in the literature. We compared a selection of models used in that work and explored the use of a replay buffer. Experiments were conducted with SlimageNet64, a cut-down version of Imagenet, and Omniglot. This work approaches instance learning from the main body of CFSL benchmarks, which utilise low-resolution, object-centred imagery. We artificially introduce distortions with noise and occlusion, which is more challenging and creates variation across observations of the same instance.

## Contribution

The main contributions are: a) enhanced CFSL framework, comparable to other continual learning benchmarks and including instance learning; b) highlighted relevance of instance learning; and c) exploration of the potential contribution of replay on CFSL tasks.

## Related work

### Few-shot learning

MAML (Model-Agnostic Meta-Learning) [8] is an early and influential Meta-Learning algorithm. Meta-learning is often described as "learning to learn", i.e., training to quickly acquire knowledge and exploit new data, while representing existing data in ways that generalise to new tasks. The paper differentiates fast acquisition of new tasks (few-shot learning) and the capability to leverage learning from previous tasks while learning new ones (meta-learning) as a solution to few-shot learning. The MAML paper also defined a popular few-shot learning benchmark that includes reinforcement learning, regression, and classification tasks, the latter using Omniglot [4] and Mini-ImageNet [5].

These classification tasks have served as a standardised set of tasks or datasets on which various few-shot and meta-learning algorithms can be evaluated and compared. However, more recent work has argued for increasing the difficulty and complexity of tasks while continuing to introduce more capable algorithms to match.

Triantafillou et al. [9] describe Omniglot and Mini-ImageNet [5] as some of the most established benchmarks in few-shot learning but consider them too homogeneous, limited to within-dataset generalisation, and ignorant of relationships between classes; for example, they note that coarse classification (e.g. dogs vs chairs) may be much easier than fine classification (e.g. dog breeds). This is a theme that our instance test, described below, takes even further.

In response, Triantafillou et al. created a Meta-Dataset for few-shot learning. The Meta-Dataset is larger—assembled from 10 pre-existing datasets, both episodic and non-episodic. In an episodic few-shot dataset, the data is organised into episodes. Each episode includes a support set and a query set. The support set contains a small number of labelled examples for each class or concept, while the query set contains unlabelled examples that need to be classified or predicted. A non-episodic dataset lacks the episodic structure and is only divided into training, validation, and testing sets, without specific support and query sets.

Two of the ten datasets have a class hierarchy, allowing coarse-class and fine-class tasks. The Meta-Dataset benchmark is parameterised in terms of Ways (the number of classes) and

Shots (the number of training examples). They explore the comparative performance of pre-training and meta-learning using several models, including Meta-learners, Prototypical Networks, Matching Networks, Relation Networks, and MAML. Their contribution, in addition to the Meta-Dataset, is Proto-MAML, a meta-learner that combines Prototypical Networks and MAML.

## Few-Shot Instance Segmentation (FSIS)

Learning instances has not been explored in a CFSL setting, but related challenges have been explored in the Few-Shot Instance Segmentation (FSIS) literature. Segmentation requires the identification of all pixels in images or video belonging to an object, despite changes in pose, viewpoint, occlusions, or illumination. The objective of FSIS is to segment an object over multiple observations. A related task, Few-shot object detection (FSOD), concerns learning slowly to detect and segment all instances of specific classes [10]. Creating segmentation training data is labour intensive, and impractical in many applications. This creates the need for few-shot segmentation; dynamically changing the set of relevant classes creates the *incremental* few-shot segmentation task, similar to the continual learning setting. Michaelis et al. [11] describe one-shot instance segmentation on the MS-COCO dataset, which is unlabelled and includes realistic, high-resolution images.

## Continual few-shot learning

Learning both continually and with few samples is a challenging problem and appears under various names in the literature, including *Continual few-shot learning (CFSL)*, *few-shot continual learning (FSCL)*, *few-shot class-incremental learning (FSCIL)* and *few-shot incremental learning (FSIL)*. Although the definitions of these terms differ slightly, they all share the same fundamental challenges and many papers do not distinguish when comparing to other methods. Likewise, in this paper, we regard them all broadly as continual few-shot learning.

The work of Antoniou et al. [6] is the basis for the enhanced benchmark proposed in this paper. They define their Continual few-shot learning (CFSL) benchmark as a series of few-shot learning tasks. In addition to the existing Omniglot dataset, they propose the SlimageNet64 dataset for this purpose, a 'slim' version of the ImageNet dataset with only 200 instances of each class at low resolution (64x64 pixels). They compare a set of popular few-shot algorithms on this dataset, including methods that pretrain a model and then fine-tune and meta-learning approaches.

Recently, there has been intense interest in continual few-shot learning. Most studies focus on vision and use variations of the standard datasets, CUB200 [12], CIFAR [13] and ImageNet [14].

One group of approaches aims to prevent large weight changes (and hence catastrophic forgetting) with various forms of regularisation. Shi et al. [15] search for a flat local minima during background training, so that fine-tuning on novel classes is not likely to cause forgetting. Gu et al. [16] adapt the network while maximising mutual information between different level feature distributions and interclass relations. Tao et al. preserve the topology of the knowledge base using a Neural Gas [17] or an elastic Hebbian graph [18]. Dong et al. [19] use an Entity Relation Graph (ERG), where the entities are exemplar feature vectors, chosen to represent the distribution of features within a class and relational knowledge is distilled from the ERG. Kukleva et al. [20] calibrate the classifier to balance performance between base and novel classes.

There are approaches that combine representations of a fixed backbone and a continually adapting model through meta-learning [21], evolved classifiers [22] or distillation [23].

Several approaches use class prototypes for classification. Usually, they add new prototypes for novel classes and adjust the feature representations and prototypes with some form of regularisation and calibration [24] or constraints based on relationships between prototypes [25]. Some prototype approaches use semantic information to align semantic and visual prototypes [26–28], which is a form of semantically driven regularisation. Semantic labels help to ensure that the structure of the latent space remains consistent. Another prototype method uses a HyperTransformer [29], which creates CNN model weights for the feature extractor, and is set up to use the previous weights as input [30].

Some of the recent papers have also used replay methods, with rehearsal [31, 32] or generative replay [33]; the latter notably used the robotic vision dataset, CORE50 [34].

Brain-inspired 'slow and fast weights' [32, 35] allow fast updates (fast weights) without overfitting (slow weights), in conjunction with a replay mechanism inspired by biological models of synaptic plasticity with memory replay [32].

It is interesting to see the incorporation of a variety of ML techniques in the discussed methods, including meta-learning [21, 25, 31, 36] adversarial training [31], active learning [33] and data augmentation [37].

All of the methods discussed so far are designed for vision. There is also recent work on CFSL in NLP [38], relation learning [39] and cybersecurity [36]. In [36], Xu et al. implemented a metric-based, meta-learning approach with fine-tuning and a memory module, which combines elements of the two baselines used in our study.

**Benchmarks.** Caccia et al. [40] propose a CFSL benchmark called OSAKA (Online Fast Adaptation and Knowledge Accumulation). OSAKA requires models to learn new out-of-distribution tasks as quickly as possible, while also remembering older tasks. Out-of-distribution shifts include replacing one dataset (such as Omniglot) with another (such as MNIST). OSAKA deliberately blurs the boundaries of episodes and instead focusses on the tasks currently being evaluated. Tasks can reoccur, and new tasks appear. Performance is measured cumulatively, during the introduction of new classes and not only after exposure has completed. The shifts between tasks are stochastic and are not observable to the model (the authors note that some continual learning methods such as EWC rely on this knowledge). The target distribution is a context-dependent, non-stationary problem.

The authors envisage that OSAKA is a very challenging benchmark. They provide a number of reference models, several based on MAML, and propose Continual-MAML, which detects and reacts to out-of-distribution data. Most methods perform badly under OSAKA conditions. More recently, some authors used the aforementioned Meta-Dataset introduced by Triantafillou et al. [9] for few-shot learning to select in and out-of-distribution tasks.

## Beyond episodic continual learning

Time-series data also motivate a move away from discrete and detectable episodes. Harrison et al. [41] sought to eliminate known and abrupt task transitions or episode segmentation. They look at time-series data, where latent task variables undergo discrete, unobservable, and stochastic changes. Observable data are dependent on latent task variables. For time-series CFSL they propose Meta-learning via Changepoint Analysis (MOCA)—a meta-learning algorithm with a changepoint detection scheme. Their benchmark has two phases, meta-learner training and online adaptation (evaluation).

Ren et al. [42] also extend few-shot learning to an online, continual and *contextual* setting, with online evaluation while learning novel classes, like OSAKA. Contextual here refers to a changing, partially observable process that affects the desired classification task. Context information is provided as a background to image classification datasets, including Omniglot and

ImageNet. The contextual task tests an agent's ability to quickly learn the meaning of a class in a specific context, thereby adapting to that change.

## Replay methods for continual learning

A variety of replay methods have been used for continual learning, surveyed in [3, 43]. Replay methods are largely inspired by Complementary Learning Systems (CLS) [44–46], a computational framework for learning in mammals. In CLS, a short-term memory (STM) representing the hippocampus stores representations of specific stimuli in a highly non-interfering manner. Interleaved replay to a long-term memory (LTM) enables improved learning and retention. The LTM is often assumed to be a model of the neocortex and is considered an iterative statistical learner of structured knowledge.

The most common replay approach is to store samples in a buffer and replay them during training. One of the challenges is memory capacity, leading to various strategies for selecting the subset of training data to store. Strategies include maximising novelty [47], using lower-dimensional latent representations [48, 49], maximising diversity [50], and selecting for an equal distribution over classes, as in NSR+ in [43]. There are also different strategies for using the buffer contents in new learning, broadly grouped into Rehearsal and Constraint methods. In rehearsal, buffer samples (typically randomly sampled) are presented for training, either in sleep phases [47], or interleaved with new training data. Constraint methods constrain the learning of new tasks using buffered samples, e.g., so that the loss does not increase with new tasks [51]. Another approach to replay, inspired by the generative nature of the hippocampus, is to generate representative samples from a probabilistic model and interleave with new tasks [52–54]. These approaches have dramatically reduced memory requirements.

## Benchmark setup

We first give an overview of the CFSL framework by Antoniou et al. [1], upon which our study is based. Second, we describe experiments to scale selected tests from Antoniou et al. [1], referred to as 'CFSL at scale'. Third, we introduce the instance test.

### Continual few-shot learning framework—Background

For context, we recap the relevant method and terminology used in the CFSL framework. In continual learning, new tasks are introduced in a stream, and old training samples are never shown again. Performance is continually assessed on new and old tasks. In the CFSL framework, new data are presented with groups of samples defined as 'support sets', and then the model must classify a set of test samples in a 'target set'. The target set contains samples from classes shown in that episode. The experiment is parameterised by a small set of 'experiment hyperparameters' described in Table 1. By varying these parameters, the experimenter can flexibly control the few-shot continual learning tasks, the total number of classes (NC), samples

**Table 1. The 'experiment hyperparameters' that fully define an experiment in the CFSL framework by Antoniou et al. [6].**

| Parameter | Description |
| --- | --- |
| NSS | Number of support sets |
| CCI | Class-change interval e.g. if CCI = 2, then the class will change every 2 support sets |
| $n$-way | Number classes per set |
| $k$-shot | Number of samples per support class in a support set |

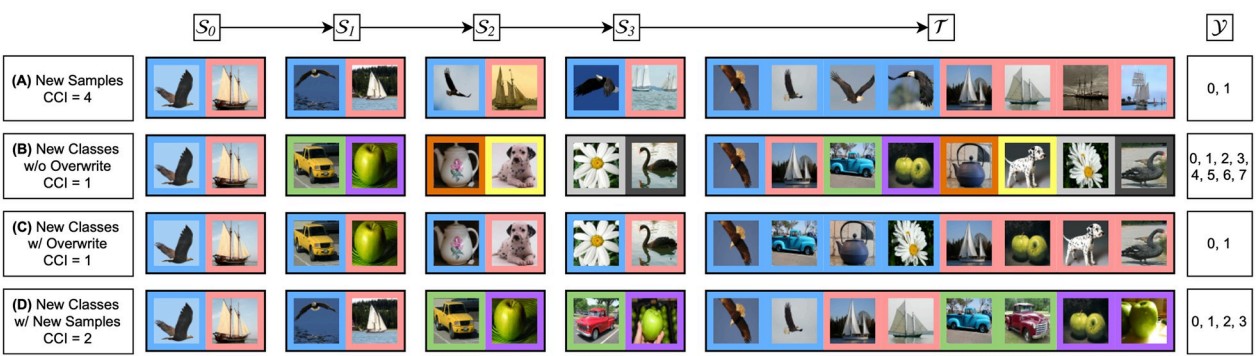

Visual representation of the four continual few-shot task types. Each row corresponds to a task with Number of Support Sets, NSS=4, and a defined Class-Change Cnterval (CCI). Given a sequence of support sets, $\mathcal{S}_n$, the aim is to correctly classify samples in the target set, $\mathcal{T}$. Colored frames correspond to the associated support set labels.

**Fig 1. Visual representation of CFSL experiment parameterisation.** Reproduced with permission from [6].

per class, and the manner in which they are presented to the learner. A visual representation is shown in Fig 1.

## CFSL at scale

In the first set of experiments, we replicated a representative set from Antoniou et al. [6], but extended the number of classes. The objective was to provide results that are more comparable to other continual learning studies in the field.

We chose to base our experiments on the parameters of Task D (see Fig 1), described in [6], as it resembles the most common and applicable real-world scenario—it introduces both new classes and additional instances of each class.

**Framework baseline—Replicating the original experiments.** During our work with CFSL, we identified and fixed several issues in the original CFSL codebase [6], and collaborated with the authors to have them reviewed and merged upstream. Given the significance of some of these issues, we opted to replicate a selected number of the original experiments (by setting the same experiment hyperparameters) to properly contextualise our new experiments and results. The main issues related to a) Pretrain+Tune weight updates and b) mislabelled new instances, which became an issue where CCI>1, see S1 File for details.

**Scaling.** In the field of continual learning, it is common for the number of classes to range from 20 to 200, even if the number of tasks in a sequence may be small (approximately 10). Therefore, we introduced experiments with up to 200 classes (compared to 5 classes per support set and a maximum of 10 support sets in [6]). We experimented with presenting the classes in two ways: **Wide**, in which the number of support sets was small but with a larger number of classes per set, and **Deep**, where there were a larger number of support sets but with a smaller number of classes per set, see Fig 2.

In a real-world setting, the way that the samples are presented, Wide vs Deep, is not directly tied to how the learner experiences new classes, but rather a choice about training method. For example, the same stream of samples could be organised as Wide or Deep. However, Wide may be more suited to scenarios where you are exposed to a wider range of new classes simultaneously, like exploring a completely new environment, whereas Deep may be better suited to learning incrementally about a narrow environment or task.

Two of the replication configurations were used as baselines, with 10 and 20 classes. Then, we created Wide and Deep configurations, with 20 total classes like the 2nd baseline. Finally,

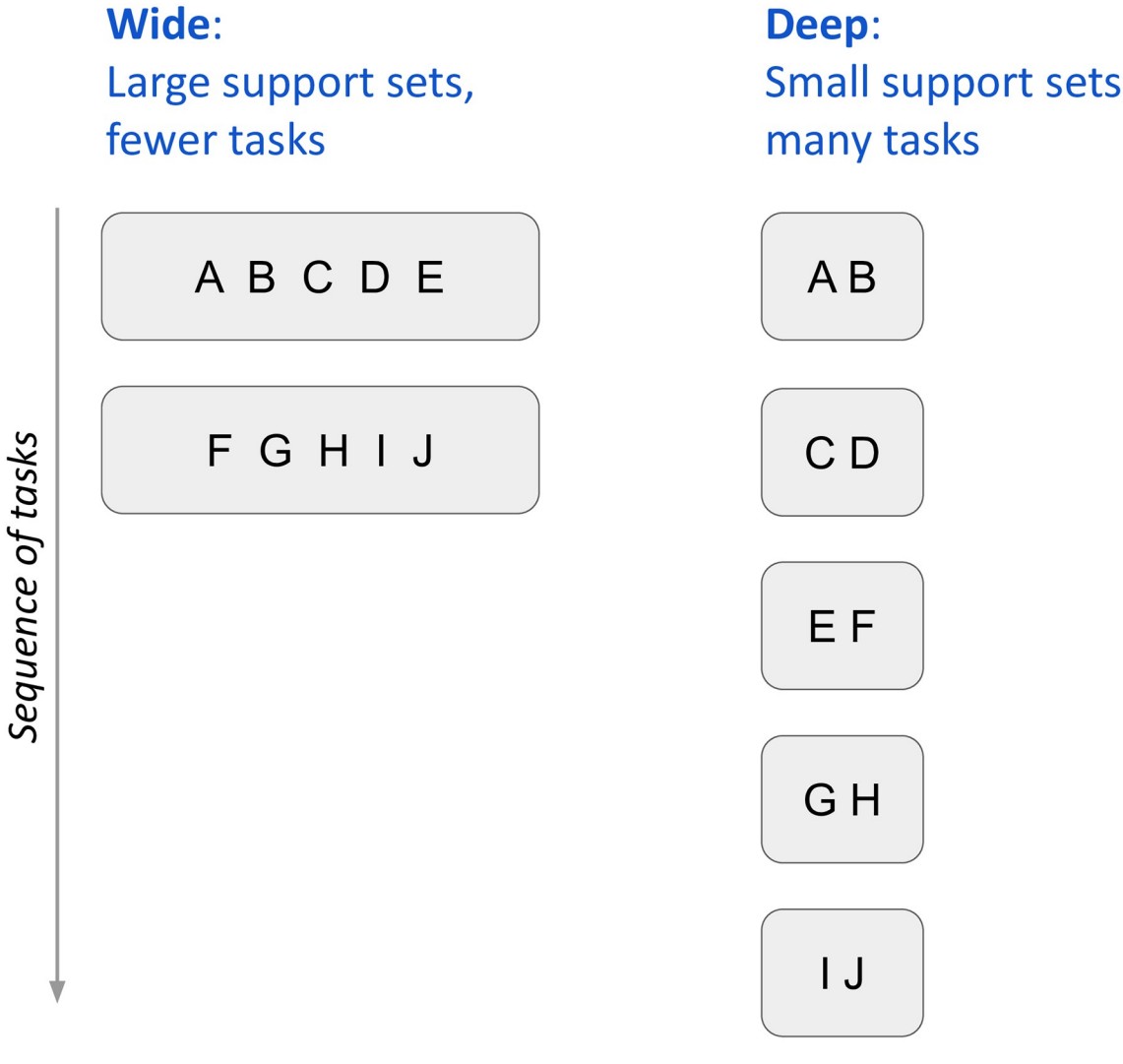

**Fig 2. Wide vs Deep.** An illustration of Wide vs Deep experiments. Wide have big support sets and few tasks, Deep has small support sets and many tasks.

the number of classes was increased ten-fold to 200, presented in both Wide and Deep configurations. In all of the experiments, $k$-shot is set to 1, so for any support set, there is only one exemplar per class. The other experiment hyperparameters (NSS, CCI and $n$-way) were chosen to achieve the desired total number of classes (NC) and the Wide and Deep configurations described above (see Fig 2 for an illustration of how they can be used to manipulate support set size and number of tasks).

### Instance test

As discussed above, we seek to increase the range of natural conditions and challenges that are captured by the CFSL benchmark, to include recognition of a specific single object—one

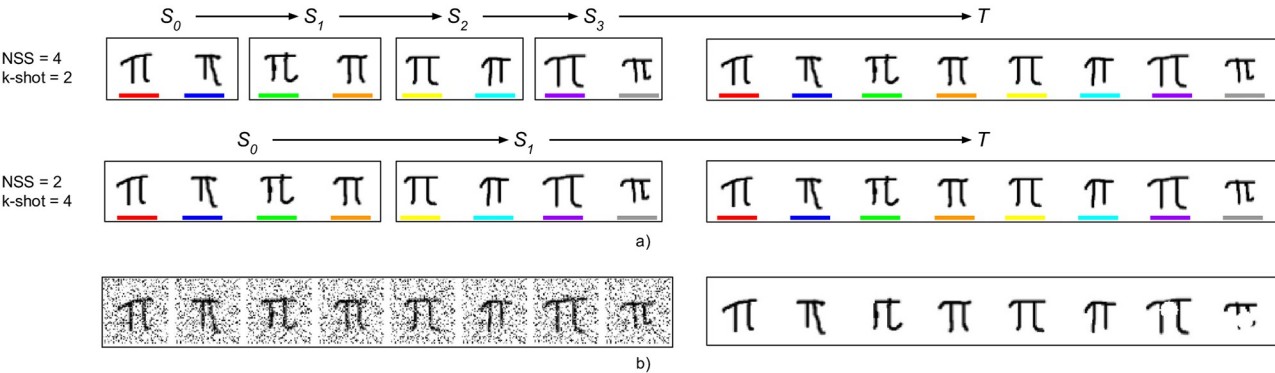

**Fig 3. Instance test:** a) Two configurations with 8 instances. In the first configuration, there are 4 support sets ($S_n$), in the second there are 2. The test set (T) consists of the same instances that were shown throughout the support sets. For each test sample, the challenge is to identify the matching identical instance from the support sets, amongst other highly similar instances. NSS = Number of support sets, and $k$-shot = the number of instances that are shown for a given class. The samples are colour coded to show which ones are identical. b) An example of images with added noise and occlusion (the fraction of pixels for noise and width of image for occlusion is 30%).

instance of a class. We call this the 'instance test', based on the instance test in AHA [55]. The learner must learn to recognise specific exemplars amongst sets where all the exemplars are drawn from the same class. An important feature of the instance test is the addition of increasing levels of corruption, intended to resemble real-world conditions and imperfect sensor readings. White noise models scenarios such as where the object is dirty, colour changed or there are varying light conditions. Occlusion models incomplete sensing, most commonly due to partial obstruction by another object.

The instance test is not simply trying to find duplicates, since the varying levels of random corruption (noise and occlusion) are applied. It is a setting that has been largely ignored in ML, and yet it represents many important real-world situations. For example, a person can easily identify their own specific cup of coffee, or their own locker, and not see them simply as a generic cup or locker class, despite those objects appearing differently due to occlusion and lighting changes. For a tangible robotics example, take a pick-and-place robot that should place items in a specific bin (rather than any generic bin).

Like in the regular tests, an episode consists of several support sets, see Fig 1. However, in the instance test, all samples are drawn from the same class and are therefore very similar to each other. The test set consists of the same instances, and the challenge is to classify each instance, i.e., which of the support set instances corresponds to a given identical instance in the test set. It is a challenging problem because they are very similar to each other. A helpful way to think of it is classification of similar classes, and each class has only 1 sample. The test is illustrated in Fig 3.

Due to the flexibility of the CFSL framework, the instance test can be implemented as a special case of the existing parameters. $n$-way is set to 1, so that there is only one class in each support set. CCI is equal to NSS so that there is no class change between support sets. Then the $k$-shot or samples per class determines how many instances are shown for a given class, which we refer to as Number of Instances, NI. We used a constant total number of instances for all experiments, 20, but experimented with presenting the samples differently, in terms of number and size of support sets. We reused empirically optimal hyperparameters from the Scaling Experiments.

For the random image corruption, noise is presented by replacing a random set of pixels, each with an intensity value, randomly sampled from a uniform distribution. Occlusion is

implemented by placing circles at random positions (and completely contained within the image). The level of corruption is denoted by a fraction, which refers to the number of pixels in the case of noise, and the proportion of the width in the case of occlusion. We used higher levels for SlimageNet64 in order to achieve the same deterioration in accuracy. Note that SlimageNet64 images have background content which can be used for recognition even with substantial noise and occlusion.

## Experimental method

In this section, we describe the training process and models, and then the experimental setup. The source code for all experiments is located at https://github.com/cerenaut/cfsl.

### Training methods and architectures

We selected three baseline methods from the original CFSL paper [6] to represent each family of algorithms tested, using the implementations published in their open source codebase (https://github.com/AntreasAntoniou/FewShotContinualLearning). The first method was Pretrain+Tune using a CNN architecture based on stacking convolutional VGG blocks [56]. The second method was Prototypical Networks (ProtoNets) [57], using the same underlying network architecture. We intended to also evaluate SCA [58], which is a complex and high performing meta-learning approach. However, we encountered resource constraints and were unable to successfully complete the larger variant of each experiment type.

To ensure a fair comparison between models and experiments with varying number of tasks/classes, we optimised hyperparmeters for each experiment, using univariate sweeps. The hyperparameter search included the architecture space: number of filters, number of VGG blocks and learning rate. Under these conditions, both ProtoNets and Pretrain+Tune methods could explore the same architectures.

The experiments were carried out on Omniglot [5] and SlimageNet64 [6] datasets. Each was split into training, validation and test sets. The SlimageNet64 splits were chosen to ensure substantial domain-shift between training and evaluation distributions to provide a strong test of generalisation [1].

All models were pre-trained until plateau (30–50 epochs for SlimageNet64 and 10 epochs for Omniglot) with 500 update steps per epoch. At the end of each epoch, the models were validated on the CFSL tasks (200 episodes consisting of support and target test sets as described above in 'Continual few-shot learning framework—backgrounds'). At the conclusion of pre-training, the best performing model was selected and tested on the CFSL tasks. Experiments were repeated 5 times with a different random seed for data sampling and model initialisation. For all the models, weights were initialised with Xavier, uniform random distribution. For pre-training, Adam Optimiser was used with cosine annealing scheduler and weight decay regularisation (value = 0.0001). For fine-tuning, the optimiser was a straightforward gradient descent optimiser without momentum.

For comparison with Antoniou et al. [1] in 'Framework baseline—replicating the original experiments', we used a 5-model ensemble. For the other experiments, we preferred to see a more direct measure of performance and used a single model.

**Pretrain+Tune.**   In this method, the model is trained on a large corpus prior to the CFSL tasks. Then during the CFSL tasks, the model is fine-tuned on the support set images before being evaluated.

The architecture consists of a VGG-based architecture with a variable number of blocks and a head with 1 dense linear layer. Like in VGG, the blocks consisted of a 2d convolutional layer with a variable number of 3x3 filters with stride and padding of 1, leaky ReLU activation

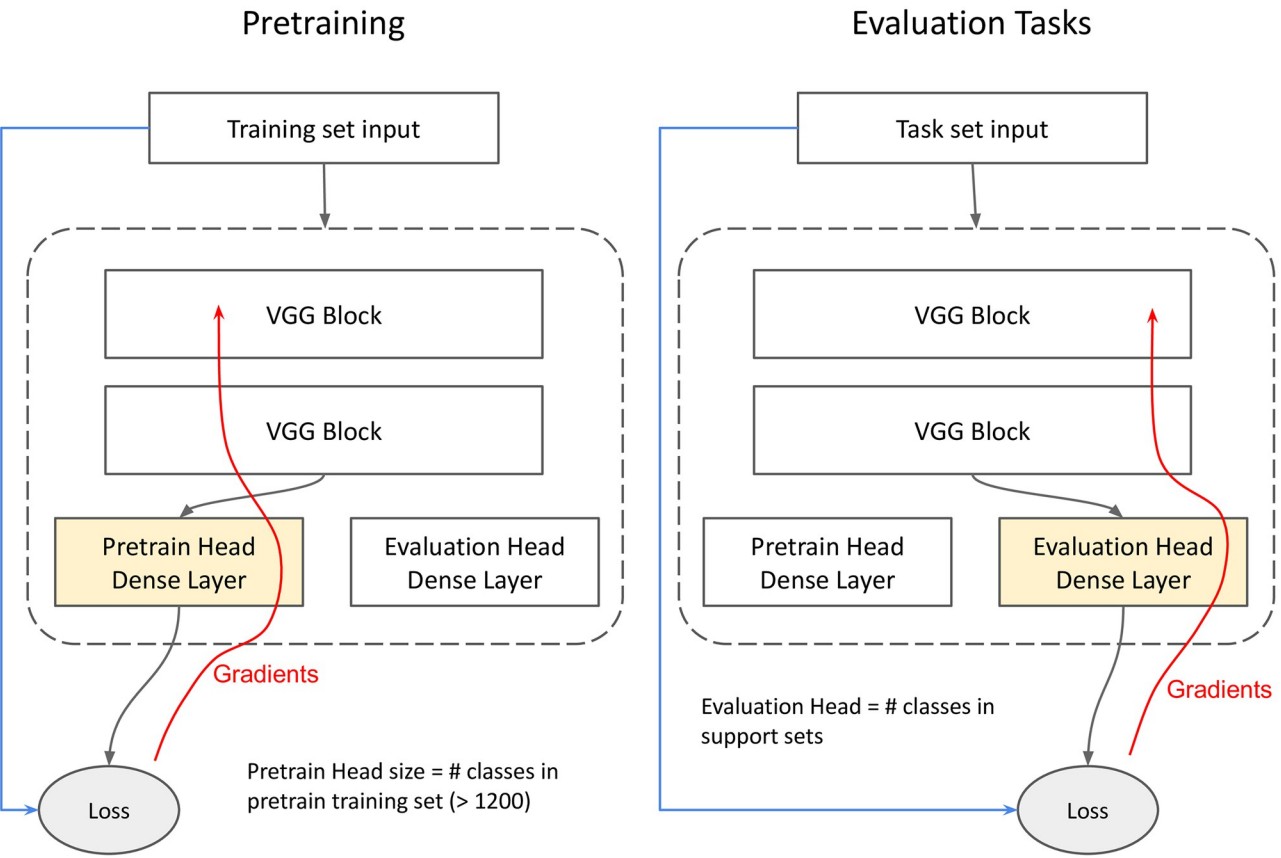

**Fig 4. Pretrain+Tune training and evaluation: Different heads are used for pretraining and each task, because the number of classes varies between pretraining and task settings.** All trainable parameters in the head and VGG blocks are adapted during pretraining and fine-tuning.

function, a batch norm layer and a max-pooling layer with stride of 2 and a 2x2 kernel. See S1 File for more details.

A distinct classification head was used for the pretraining phase, with the same number of classes as the pretraining set. At the end of pretraining, the top $q$ models were chosen using the validation set. $q = 5$ for the ensembling approach used in the Replication experiments, and 1 for all the others. Then in evaluation, for each task, all the weights were reset to the pretrained values and a newly initialised head was used. Fine-tuning adjusted the weights throughout the network, including the VGG blocks as well as the classifier head. The process is illustrated in Fig 4.

**ProtoNets.** Prototypical Networks (ProtoNets) train a network to learn an embedding that is optimised for matching [57]. The embedding for each class is considered to be a prototype for that class. In these experiments, the architecture details and hyperparameterisations are very similar to Pretrain+Tune above, except that no bias was used and no dense layer was required (as the learning objective is calculated from the embedding without the need for a standard classifier).

The same dataset splits were used as for Pretrain+Tune. In this case, the pretraining split was used to learn the embedding, and the evaluation split was used for the tasks in the same way. Cosine similarity was used to calculate the distance between embeddings to perform classification. No learning occurred during the evaluation tasks.

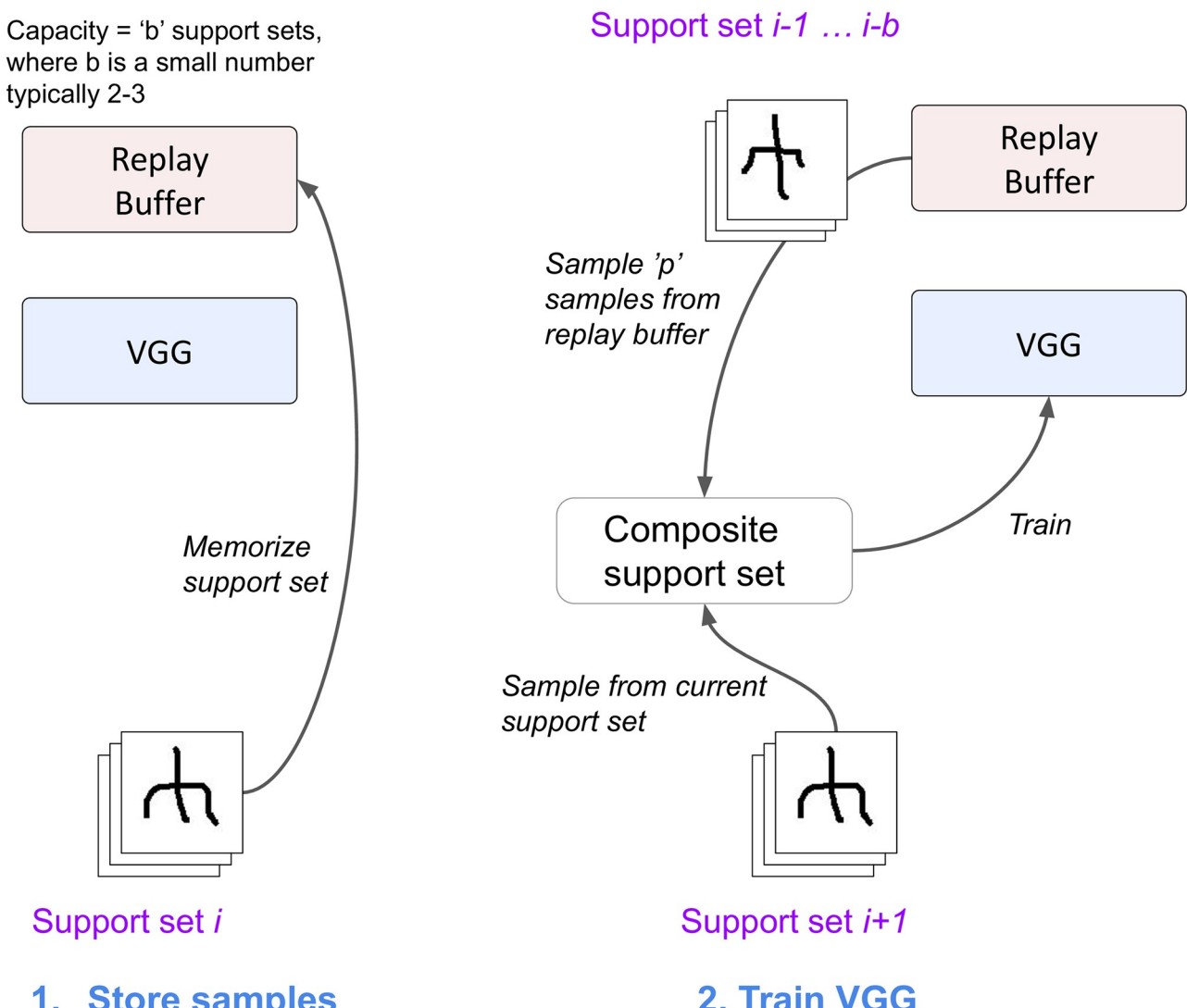

**Fig 5. Learning with replay.** Complementary Learning Systems (CLS) setup with Long Term Memory (LTM), paired with a circular buffer Short Term Memory (STM). First, in a memorisation step, the STM temporarily stores recent support sets. Second, in a recall step, the memorised data are used in LTM training.

**Learning with replay.** As described in 'Related work', replay methods have been applied to continual learning [3, 43], and more recently to continual few-shot learning [31, 32]. In this work, we created a very simple replay mechanism to provide initial exploration of the performance benefit of replay methods. It was applied to the Pretrain+Tune method and is referred to as Pretrain+Tune+Replay.

The Pretrain+Tune+Replay architecture is shown in Fig 5. The long-term memory (VGG network) is augmented with a short-term memory (STM), consisting of a circular buffer, in which new memories replace older memories in a FIFO (first in, first out) manner. The STM stores samples from recent tasks and replays them by interleaving samples from the STM with samples from current tasks during fine-tuning. The process is divided into two stages. First, the current support set is stored in the STM, adding to recent support sets. *b* is the buffer size,

measured in support sets, and is a tuneable hyperparameter. Second, the network is trained using the current support set, as well as samples randomly drawn from the replay buffer. The number of samples is determined by a second hyperparameter $p$. The hyperparameter search was expanded to include both $b$ and $p$ in the replay experiments.

In this initial exploration of replay for CFSL, we applied it only to the Pretrain+Tune method. Replay fits naturally, as weights are already adapted during fine-tuning and it follows the precedent of other replay methods in the literature (see 'Establishing benchmarks in continual and few-shot learning'). In contrast, the conceptual approach of ProtoNets is to meta-learn a fixed embedding using a wide spread of classes from a background data set and generalise well from that, without further modifications. Replay for ProtoNets would be an interesting challenge for future work.

## Experimental setup

The code was written using the PyTorch framework v1.6.0. The Omniglot experiments were conducted on two machines. Machine 1 had a GeForce GTX 1070 GPU with 8GB RAM and an Intel Core i7–7700 with 32GB RAM. Machine 2 had a GeForce GTX 1060 GPU with 6GB RAM and an Intel Core i7–7700 with 16GB RAM. The SlimageNet64 experiments were conducted on cloud compute using virtual machines with an A10 GPU with 24GB and 30 vCPUs, 200GB RAM, using Lambda Labs https://lambdalabs.com/. The duration of training for an individual experiment was in the order of half a day, including pretraining and multiple seeds.

## Results

### CFSL at scale

**Framework baseline—Replicating the original experiments.** The results of the replication experiments are summarised in Table 2, which includes reference values from Antoniou et al. [6] for comparison. In experiments that were affected by code fixes (Pretrain+Tune weight updates and where CCI>1), performance improved substantially from unusually low values, and performance across experiments followed a more expected trend (i.e., increasing accuracy of Pretrain+Tune with decreasing number of classes). ProtoNets was substantially more accurate than Pretrain+Tune, and performed consistently across different variations of the presentation of 10–50 total classes.

**Table 2. Replication experiments.** Replication of Task D from [6] after correcting errors in the framework code. Accuracy is shown in %, as mean ± standard deviation across 3 random seeds.

| Method name | NSS | CCI | *n*-way | *k*-shot | Number of classes | Ensemble accuracy | Accuracy | Reference ensemble accuracy |
|---|---|---|---|---|---|---|---|---|
| **Pretrain+Tune** | 4 | 2 | 5 | 2 | 10 | 37.81 ± 0.77 | 36.7 ± 0.80 | 7.91 ± 0.15 |
| **Pretrain+Tune** | 8 | 2 | 5 | 2 | 20 | 27.92 ± 0.10 | 26.41 ± 0.14 | 3.86 ± 0.06 |
| **Pretrain+Tune** | 3 | 1 | 5 | 2 | 15 | 17.76 ± 0.32 | 17.40 ± 0.33 | 9.97 ± 0.14 |
| **Pretrain+Tune** | 5 | 1 | 5 | 2 | 25 | 13.76 ± 0.08 | 13.10 ± 0.03 | 6.02 ± 0.02 |
| **Pretrain+Tune** | 10 | 1 | 5 | 2 | 50 | 9.73 ± 0.06 | 8.36 ± 0.05 | 3.13 ± 0.03 |
| **ProtoNets** | 4 | 2 | 5 | 2 | 10 | 97.93 ± 0.05 | 96.98 ± 0.05 | 48.98 ± 0.03 |
| **ProtoNets** | 8 | 2 | 5 | 2 | 20 | 96.66 ± 0.03 | 95.22 ± 0.06 | 48.44 ± 0.03 |
| **ProtoNets** | 3 | 1 | 5 | 2 | 15 | 97.12 ± 0.06 | 95.88 ± 0.12 | 95.30 ± 0.12 |
| **ProtoNets** | 5 | 1 | 5 | 2 | 25 | 95.93 ± 0.12 | 94.36 ± 0.05 | 91.52 ± 0.20 |
| **ProtoNets** | 10 | 1 | 5 | 2 | 50 | 92.43 ± 0.27 | 90.24 ± 0.10 | 83.72 ± 0.19 |

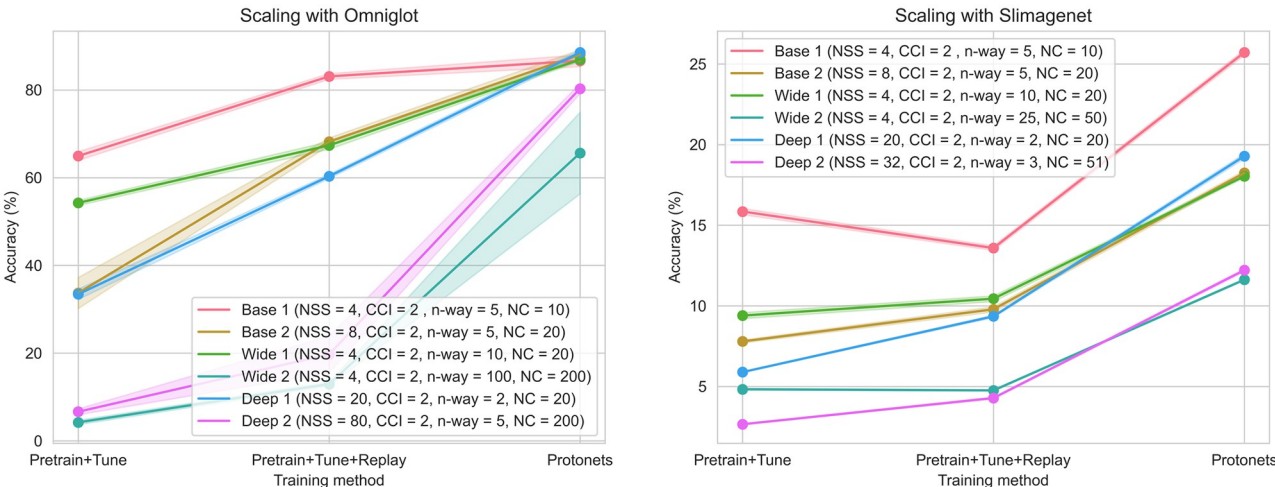

**Fig 6. Scaling experiments.** In the CFSL scaling experiments, the number of classes was increased to 200. Three approaches were compared: i. Pretrained network with fine-tuning, ii. Pretrained network with fine-tuning and the addition of Replay, and iii. ProtoNets. The bold line shows the mean, and the shaded area shows one standard deviation across 5 random seeds.

**Scaling—Omniglot.**  The results are summarised in Fig 6(a). See S2 File for accuracies in tabular format, hyperparameters and number of fine-tuning training steps for the replay experiments.

Overall, increasing the number of classes from 20 to 200 led to a dramatic decrease in accuracy. The manner in which the classes were presented ('Wide' or 'Deep') affects performance substantially. The ProtoNets method had the best accuracy. The Pretrain+Tune method was substantially improved by the addition of Replay, reaching a performance similar to that of ProtoNets in Baseline 1.

**Scaling—SlimageNet64.**  Results for SlimageNet64 experiments are summarised in Fig 6 (b). See S2 File for accuracies in tabular format, hyperparameters and number of fine-tuning training steps for the replay experiments. Overall, accuracy is lower than for Omniglot images; ProtoNets delivered the best accuracy. The benefit of Replay was less dramatic, but still present in all settings except Baseline 1.

## Instance test

The results for the instance test are summarised in Table 3 (Omniglot), Table 4 (SlimageNet64) and Fig 7. The number of 'items to identify', which in this case are separate instances, is constant at 20 for all of the experiments. Five experimental settings were tested, varying parameters NSS, *k*-shot and NI.

Model hyperparameters from Baseline 2 were reused due to the similarity of the setting, in lieu of further hyperparameter search. The number of fine-tuning training steps for the replay experiments are given in Section 3, S2 File.

Compared to classification experiments, accuracy is relatively high for ProtoNets. The Pretrain+Tune method is noticeably worse than ProtoNets, and as observed in classification experiments, Replay provides a substantial improvement in accuracy, but in most experiments does not provide accuracy comparable to ProtoNets.

**Table 3. Instance test: Omniglot.** Accuracy is shown in %, as mean ± standard deviation across 5 random seeds. $n$-way = 1 for all experiments, to restrict distinguishing between similar instances of a single class. In the instance test, $k$-shot translates to the size of the support set. It is 1-shot in the sense that each instance is only shown once. NI, number of instances = 20 for all the experiments.

| Method name | Exp. 1 NSS = 2, $k$-shot = 10, NI = 20 | Exp. 2 NSS = 4, $k$-shot = 5, NI = 20 | Exp. 3 NSS = 10, $k$-shot = 2, NI = 20 | Exp. 4 NSS = 20, $k$-shot = 1, NI = 20 |
|---|---|---|---|---|
| Pretrain+Tune | 40.35 ± 1.40 | 47.63 ± 5.00 | 47.72 ± 2.77 | 43.88 ± 2.47 |
| Pretrain+Tune +Replay | 96.39 ± 1.36 | 89.19 ± 2.79 | 82.77 ± 2.15 | 79.46 ± 3.01 |
| ProtoNets | 92.38 ± 1.00 | 91.94 ± 1.09 | 91.79 ± 1.16 | 91.79 ± 1.16 |

**Table 4. Instance test: SlimageNet64.** Accuracy is shown in %, as mean ± standard deviation across 5 random seeds. $n$-way = 1 for all experiments, to restrict distinguishing between similar instances of a single class. In the instance test, $k$-shot translates to the size of the support set. It is 1-shot in the sense that each instance is only shown once. NI, number of instances = 20 for all the experiments.

| Method name | Exp. 1 NSS = 2, $k$-shot = 10, NI = 20 | Exp. 2 NSS = 4, $k$-shot = 5, NI = 20 | Exp. 3 NSS = 10, $k$-shot = 2, NI = 20 | Exp. 4 NSS = 20, $k$-shot = 1, NI = 20 |
|---|---|---|---|---|
| Pretrain+Tune | 44.32 ± 0.27 | 23.23 ± 0.30 | 13.62 ± 0.18 | 6.73 ± 0.07 |
| Pretrain+Tune +Replay | 69.98 ± 0.18 | 63.49 ± 0.45 | 71.29 ± 0.44 | 66.88 ± 0.22 |
| ProtoNets | 99.72 ± 0.07 | 99.77 ± 0.05 | 99.76 ± 0.05 | 99.78 ± 0.06 |

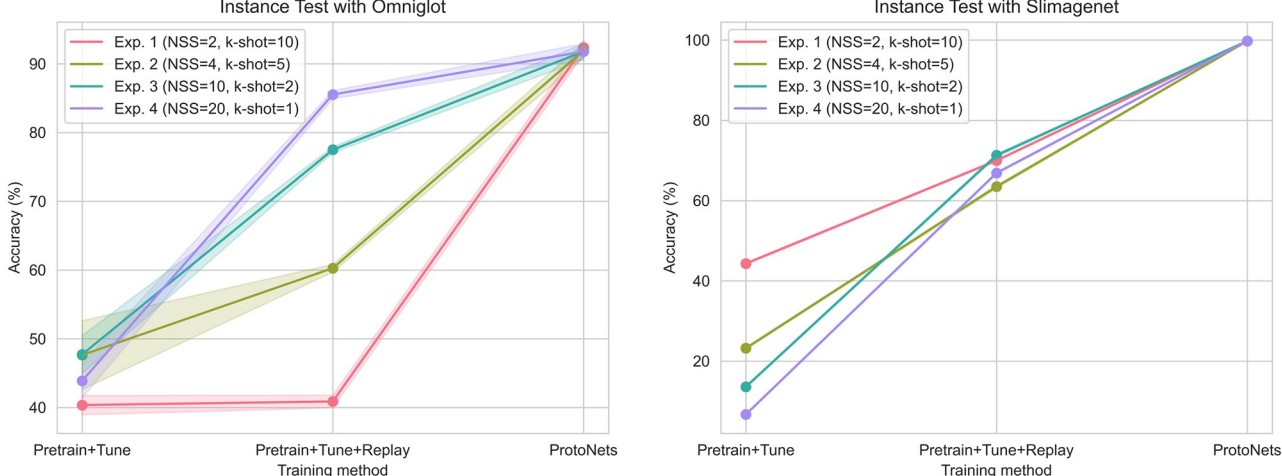

**Fig 7. Instance test.** In the instance test, $k$-shot translates to the size of the support set. It is 1-shot in the sense that each instance is only shown once. NI, number of instances = 20 for all the experiments. The bold line shows the mean, and the shaded area shows one standard deviation, across 5 random seeds.

## Instance test—With noise and occlusion

Figs 8 and 9 illustrate the results of the three methods in Instance Test Experiments 1–4 under varying levels of noise and occlusion. Details of experiment configurations can be found in Tables 3 and 4.

With smaller amounts of noise and occlusion, the results and in particular the ranking of the three methods are unchanged. However, with larger amounts of noise and occlusion, the Pretrain+Tune+Replay method is often more accurate than ProtoNets on both image datasets.

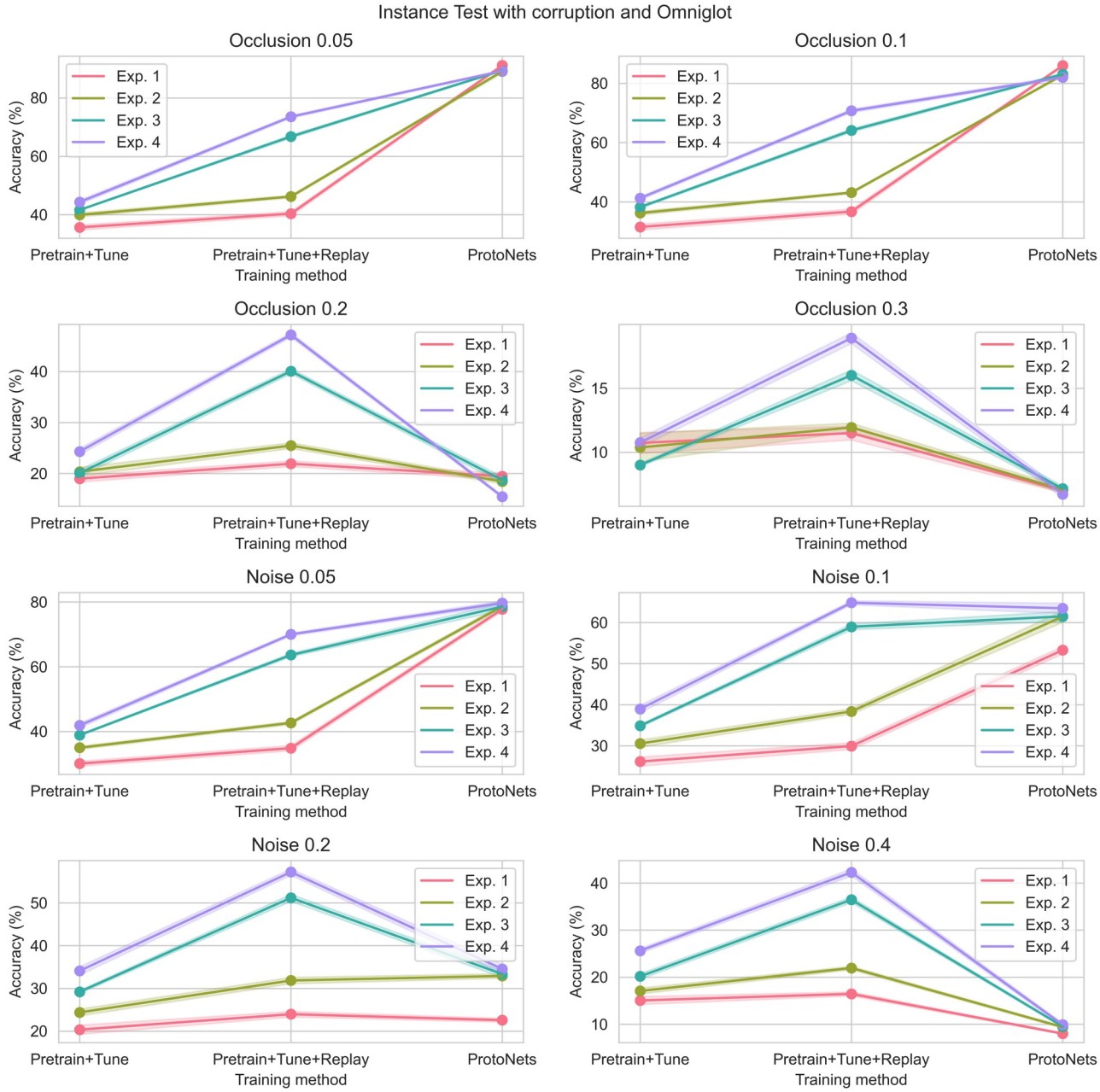

**Fig 8. Instance test: Omniglot images, with noise or occlusion.** Refer to earlier figures and tables for the definition of each experiment.

## Discussion

In this study we found that for the methods tested, few-shot continual learning of new classes is more difficult at scale, i.e., as the number of classes was increased from 20 to 200. Performance of all methods in the novel instance test was comparable to performance on similarly sized classification tasks at scale.

The ProtoNets method outperformed the Pretrain+Tune method in all tasks. With the addition of replay, Pretrain+Tune accuracy improved substantially, becoming comparable to

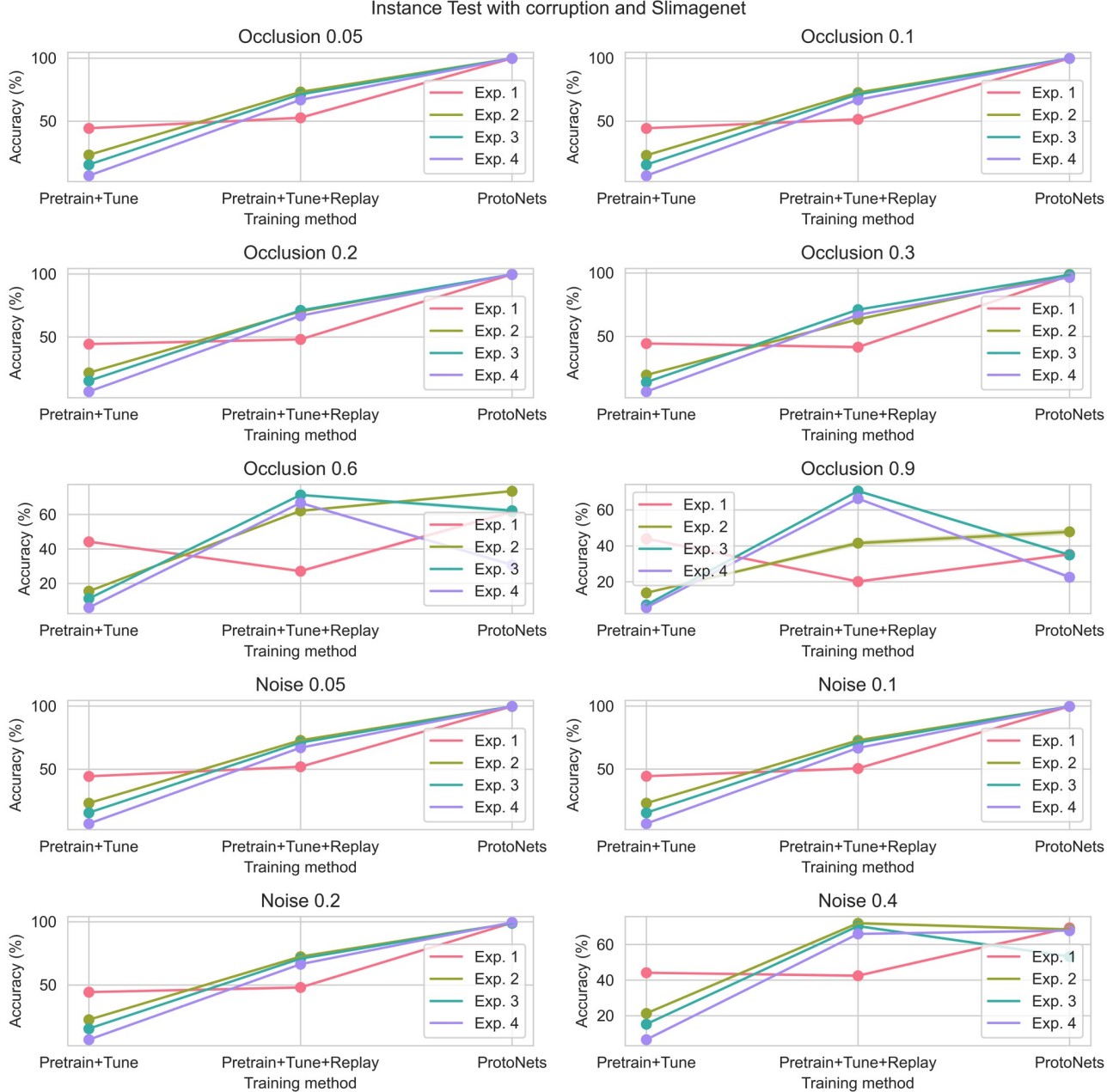

**Fig 9. Instance test: SlimageNet64 images, with noise or occlusion.** Refer to earlier figures and tables for the definition of each experiment.

ProtoNets in some settings. With higher levels of noise and occlusion in the instance test, Pretrain+Tune+Replay is often superior to ProtoNets, demonstrating that these conditions evaluate different capabilities.

## Comparison of Pretrain+Tune and ProtoNets methods

The experiments involved two base methods: Pretrain+Tune and ProtoNets [57]. It is natural to compare their performance, but comparison should be cautious as ProtoNets and Pretrain+Tune methods do not perform the same learning task.

In ProtoNets, classification is achieved by comparing embeddings, which requires a short-term memory (STM) of the reference embedding being matched. By convention, that memory is in the testing framework rather than in the ProtoNet architecture itself. Given this perspective, adding the STM (replay buffer) to the Pretrain+Tune network makes it conceptually more similar to ProtoNets, and the results are also more similar. In this study, the STM is used for replay only. In future work, it could also be used for classification for recent samples still in short-term memory, as was done in AHA [59].

Another way in which Pretrain+Tune and ProtoNet learning differs, is that ProtoNets do not actually acquire new knowledge during training and therefore do not continually learn. There is just one optimiser (for the meta-learning 'outer loop') that gets triggered during the 'pretraining' phase, meaning that it learns meta-parameters for an embedding representation that will be optimal for downstream tasks.

## Wide vs Deep—Big batches vs many tasks (no replay)

The way that data are presented, not just the number classes, makes a difference to learning. When the number of classes was held constant at 20, but the number and size of support sets was varied (Baseline 2 vs Wide 1 or Deep 1), performance was better for the wide configuration. However, when the number of classes was increased by an order of magnitude to 200 (Wide 2 vs Deep 2), performance was better in the deep configuration, where the classes were spread out across smaller batches. This was unexpected given that weight updates (in Pretrain +Tune) occur after each support set, and the more support sets there are, the more it could 'forget' earlier learning. It is possible that as the number of classes increase, the larger support sets (in the Wide experiments) are harder to learn, or cause sharper forgetting by virtue of the fact that more knowledge is being acquired in one update.

ProtoNets [57] are very effective in the scaling test, despite not actually learning during these tasks. This implies that an effective embedding space was learned during pretraining for the task. Since no learning takes place, performance cannot suffer due to forgetting. Therefore, lower performance when there are a lot of classes, as in Wide 2 and Deep 2, is likely due to learning similar embeddings for different classes.

## Specific instances (no replay)

The results suggest that one-shot distinguishing of specific (very similar) instances is not more difficult than classification, for these methods and architectures.

The fact that Pretrain+Tune accuracy increases as the instances are distributed over more support sets, further hints that CNNs may be more effective at continual learning with smaller support sets, which is in-line with our interpretation of why Deep (more, smaller support sets) was easier than Wide in the scaling experiments (explained in more detail in 'Wide vs Deep— big batches vs many tasks (no replay)' above).

ProtoNets is effective in both the instance test and classification. In the instance test, generalisation is not required, and so representations are less likely to overlap, reducing the possibility of clashes. In addition, no fine-tuning occurs (see earlier in the Discussion 'Comparison of Pretrain+Tune and ProtoNets methods'), so performance is very stable across all configurations. However, with the addition of noise or occlusion, this inflexibility significantly reduced ProtoNet recognition performance, especially for the Omniglot dataset.

ProtoNets is less disrupted by noise and occlusion in the SlimageNet64 dataset, probably because these images have background content which provides stable features that allow recognition. The desirability of using background features for recognition depends on the purpose

of the task. In some cases, recognising the scene where an object was placed might be the only way to distinguish it.

### Effect of replay

As hypothesised, adding replay to Pretrain+Tune enabled a strong improvement across tasks. Despite the improvement, performance did not reach the same level as ProtoNets in most scaling tests (classification). Replay had a more substantial impact in the instance test. In instance test conditions with higher levels of noise or occlusion, where the ProtoNets method was less robust, the Pretrain+Tune+Replay method often outperformed ProtoNets.

Although Pretrain+Tune+Replay did not convincingly outperform ProtoNets in many experiments, there are implications for future work. Replay does improve the performance of a statistical learner (i.e., an LTM) in few-shot continual learning.

Finally, and perhaps most significantly, ProtoNets as implemented do not acquire new knowledge during training, as explained earlier in this section, and therefore do not actually demonstrate continual learning. The inability to adapt is very likely to limit performance if there is a shift in the statistics of data distribution, or out of domain (e.g., as in OSAKA experiments [40]). Pushing these limits and exploring weight adaptation during training in the context of CFSL is an important area for future work.

### Limitations and future work

The base CFSL framework that we used measures performance after all the learning has occurred. In contrast, most studies in the continual learning literature document progressive performance as new tasks are introduced, and these task changes might not be observable (e.g. [41]).

The instance test described and evaluated in this paper is limited by the sophistication and realism of the images used and the limited challenge of synthetic noise and occlusion. In SlimageNet64 experiments, the background also provides a strong cue as to the correct figure match. More realistic high-resolution video imagery would probably better capture the conflicting challenges of balancing memorisation-recognition and class-generalisation.

Despite these limitations, we observed that under greater noise and occlusion conditions, neither ProtoNets nor Pretrained methods with or without Replay were satisfactory in the Instance test. It is possible that use of more sophisticated architectures could help, comprising a promising direction for future work. Firstly, more recent architectures could be used for feature extraction, such as ResNet [60] and EfficientNet variants [61]. Secondly, implementing more advanced replay mechanisms as described in related work 'Related wor', such as selective storing and retrieval of memories into the buffer, or implementing the CLS [44–46] concept of dual pathways for pattern separation and generalisation as in AHA [59, 62]. Finally, the success of ProtoNet training implies that ProtoNet+Replay would be a promising direction.

### Conclusion

Continual few-shot learning of both classes and instances is a necessary capability for agents operating in unfamiliar and changing environments. This study is one of the first steps in that direction, combining continual and few-shot learning, additionally evaluating the ability to recognise specific instances, and in doing so demonstrating that replay is a competitive approach under certain conditions.

We aimed to enhance the CFSL framework [6] and evaluate a set of common CFSL approaches on the resulting tasks. The CFSL framework was scaled to 200 classes, to make it more comparable to typical continual learning experiments. We introduced two variants,

**Wide** with fewer larger training 'support sets' and **Deep** with a greater number of smaller support sets. We also expanded the CFSL framework with a few-shot continual instance-recognition test, which measures a capability important in everyday life, but often neglected in Machine Learning.

We found that increasing the number of classes decreased classification performance (scaling test) and the way that the data were presented did affect accuracy. Performance in the few-shot instance test was comparable to few-shot classification, but results were significantly worse with the addition of basic challenges such as noise and occlusion.

Augmenting models with a replay mechanism substantially improved performance in most experiments. ProtoNet training was superior to pretraining with fine-tuning under most settings, with the exception of the instance test given high levels of noise or occlusion. This demonstrates that the instance test requires model qualities that are not evaluated under existing CFSL experimental regimes, and that in some of these conditions an LTM plus replay architecture may be preferable.

## Supporting information

**S1 File. Supplementary materials relating to the method, including CFSL framework baseline bugfixes and the Pretrain+Tune method.**
(PDF)

**S2 File. Supplementary materials relating to the results, including the number of fine-tuning steps used and accuracies in tabular format.**
(PDF)

## Author Contributions

**Conceptualization:** Gideon Kowadlo, David Rawlinson.

**Formal analysis:** Gideon Kowadlo, Abdelrahman Ahmed, David Rawlinson.

**Funding acquisition:** Gideon Kowadlo.

**Investigation:** Gideon Kowadlo, Abdelrahman Ahmed, Amir Mayan, David Rawlinson.

**Methodology:** Gideon Kowadlo, Abdelrahman Ahmed.

**Project administration:** Gideon Kowadlo.

**Resources:** Gideon Kowadlo.

**Software:** Gideon Kowadlo, Abdelrahman Ahmed, Amir Mayan.

**Supervision:** Gideon Kowadlo, Abdelrahman Ahmed.

**Validation:** Gideon Kowadlo.

**Visualization:** Gideon Kowadlo.

**Writing – original draft:** Gideon Kowadlo, David Rawlinson.

**Writing – review & editing:** Gideon Kowadlo, Abdelrahman Ahmed, David Rawlinson.

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
