## [Decision Letter · Decision Letter 0]

2 Apr 2024

PONE-D-24-03227Expanding continual few-shot learning benchmarks to include recognition of specific instancesPLOS ONE

Dear Dr. Kowadlo,

Thank you for submitting your manuscript to PLOS ONE. After careful consideration, we feel that it has merit but does not fully meet PLOS ONE’s publication criteria as it currently stands. Therefore, we invite you to submit a revised version of the manuscript that addresses the points raised during the review process.

**ACADEMIC EDITOR: **

**The authors are requested to revise the paper as per the reviewer comments and resubmit it to the journal on or before deadline. **

We look forward to receiving your revised manuscript.

Kind regards,

T. Ganesh Kumar, PhD

Academic Editor

PLOS ONE

“This initiative was partly funded by the Department of Defence and the Office of National Intelligence under the AI for Decision Making Program, delivered in partnership with the Defence Science Institute in Victoria”

“Recipient: The organisation, Cerenaut

Funder: Department of Defence and the Office of National Intelligence under the AI for Decision Making Program, delivered in partnership with the Defence Science Institute in Victoria

Grant number: N/A

URL: https://www.dst.defence.gov.au/partner-with-us/opportunities/artificial-intelligence-decision-making-initiative

The funders did not play any role in the study design, data collection and analysis, decision to publish or preparation of the manuscript.”

4. We note that Figure 1 in your submission contain copyrighted images. All PLOS content is published under the Creative Commons Attribution License (CC BY 4.0), which means that the manuscript, images, and Supporting Information files will be freely available online, and any third party is permitted to access, download, copy, distribute, and use these materials in any way, even commercially, with proper attribution. For more information, see our copyright guidelines: http://journals.plos.org/plosone/s/licenses-and-copyright.

Additional Editor Comments:

The authors are requested to revised the paper as per the reviewer comments.

Reviewers' comments:

Reviewer's Responses to Questions

**Comments to the Author**

1. Is the manuscript technically sound, and do the data support the conclusions?

Reviewer #1: Yes

Reviewer #2: Yes

Reviewer #3: Yes

2. Has the statistical analysis been performed appropriately and rigorously? 

Reviewer #1: Yes

Reviewer #2: Yes

Reviewer #3: Yes

3. Have the authors made all data underlying the findings in their manuscript fully available?

Reviewer #1: Yes

Reviewer #2: Yes

Reviewer #3: Yes

4. Is the manuscript presented in an intelligible fashion and written in standard English?

Reviewer #1: Yes

Reviewer #2: Yes

Reviewer #3: Yes

5. Review Comments to the Author

Reviewer #1: The manuscript was demonstrating Continual learning and few-shot learning in progress towards broader Machine Learning (ML) capabilities. the work shows the efficiency of the model and was presented in a proper way.

Reviewer #2: 1. Can you elaborate on how the parameters (NSS, CCI, n-way, k-shot) are chosen for each experiment?

2. Could you provide more details on the issues identified in the original CFSL codebase and how they were addressed?

3. Can you elaborate on how the instance test was designed to reflect real-world challenges, such as noise and occlusion?

4. Are there any insights into why ProtoNets were less disrupted by noise and occlusion in the SlimageNet64 dataset compared to the Omniglot dataset?

5. It is essential to ensure that the literature review encompasses the most recent advancements in the field.

Reviewer #3: The authors introduced the CFSL in two ways that capture a broader range of challenges, important for intelligent agent behaviour in realworld conditions such as by increasing the number of classes by an order of magnitude, making the results more comparable to standard continual learning experiments and introduce an ‘instance test’ which requires recognition of specific instances of classes – a capability of animal cognition that is usually

neglected in ML. The author organized the paper well.

6. PLOS authors have the option to publish the peer review history of their article (what does this mean?). If published, this will include your full peer review and any attached files.

Reviewer #1: **Yes: **JAGANNATHAN J

Reviewer #2: No

Reviewer #3: **Yes: **Poongothai Elango

---

## [Author Response · Author response to Decision Letter 0]

17 May 2024

Thanks for the review and comments. I attached a document with all review comments and our responses, it is called 'Response to Reviewers.pdf'

---

## [Decision Letter · Decision Letter 1]

6 Jun 2024

Expanding continual few-shot learning benchmarks to include recognition of specific instances

PONE-D-24-03227R1

Dear Dr. Kowadlo,

We’re pleased to inform you that your manuscript has been judged scientifically suitable for publication and will be formally accepted for publication once it meets all outstanding technical requirements.

Kind regards,

T. Ganesh Kumar, PhD

Academic Editor

PLOS ONE

Additional Editor Comments (optional):

The auhtors have incorporated the reviewer comments in the revised version.

As per the second review report, the both reviewers have accepted the paper and given the positive comments.

I have given acceptance for this paper for further process.

Reviewers' comments:

Reviewer's Responses to Questions

**Comments to the Author**

1. If the authors have adequately addressed your comments raised in a previous round of review and you feel that this manuscript is now acceptable for publication, you may indicate that here to bypass the “Comments to the Author” section, enter your conflict of interest statement in the “Confidential to Editor” section, and submit your "Accept" recommendation.

Reviewer #2: All comments have been addressed

Reviewer #3: All comments have been addressed

2. Is the manuscript technically sound, and do the data support the conclusions?

Reviewer #2: Yes

Reviewer #3: Yes

3. Has the statistical analysis been performed appropriately and rigorously? 

Reviewer #2: Yes

Reviewer #3: Yes

4. Have the authors made all data underlying the findings in their manuscript fully available?

Reviewer #2: Yes

Reviewer #3: Yes

5. Is the manuscript presented in an intelligible fashion and written in standard English?

Reviewer #2: Yes

Reviewer #3: Yes

6. Review Comments to the Author

Reviewer #2: (No Response)

Reviewer #3: (No Response)

7. PLOS authors have the option to publish the peer review history of their article (what does this mean?). If published, this will include your full peer review and any attached files.

Reviewer #2: **Yes: **M.Asha Paul

Reviewer #3: **Yes: **Dr.E. Poongothai

---

## [Editor Report · Acceptance letter]

25 Jun 2024

PONE-D-24-03227R1 

PLOS ONE

Dear Dr. Kowadlo, 

I'm pleased to inform you that your manuscript has been deemed suitable for publication in PLOS ONE. Congratulations! Your manuscript is now being handed over to our production team.

Kind regards, 

on behalf of

Dr. T. Ganesh Kumar 

Academic Editor

PLOS ONE